# Structure of Complex Concentrated Alloys Derived from Iron Aluminide Fe_3_Al

**DOI:** 10.3390/ma16155388

**Published:** 2023-07-31

**Authors:** Josef Pešička, Petr Kratochvíl, Robert Král, Jozef Veselý, Eliška Jača, Dalibor Preisler, Stanislav Daniš, Peter Minárik, Libor Čamek

**Affiliations:** 1Department of Physics of Materials, Faculty of Mathematics and Physics, Charles University, Ke Karlovu 5, 12116 Prague, Czech Republic; pekrat@met.mff.cuni.cz (P.K.);; 2Department of Condensed Matter Physics, Faculty of Mathematics and Physics, Charles University, Ke Karlovu 5, 12116 Prague, Czech Republic; 3Department of Foundry Engineering, Brno University of Technology, Technická 2896/2, 61669 Brno, Czech Republic

**Keywords:** entropy alloys, microstructure, Calphad

## Abstract

The phase structure and composition of a series of four alloys based on Fe_3_Al was investigated by means of scanning electron microscopy, X-ray diffraction and transmission electron microscopy. The materials were composed of Fe and Al with a fixed ratio of 3:1 alloyed with V, Cr and Ni at 8, 12, 15 and 20 at. % each (composition formula: Fe_3(100−3x)/4_ Al_(100−3x)/4_V_x_Cr_x_Ni_x_). For 8% alloying, the material is single-phase D0_3_. Furthermore, 12 and 15% alloying results in bcc–B2 phase separation on two length scales. Moreover, 20% alloying gives rise to the FeNiCrV σ phase supplemented by B2. These findings are discussed with respect to the results obtained via Calphad modeling using the TCHEA5 database and can serve in further improvement.

## 1. Introduction

For years, iron-aluminide-based alloys have been known as materials for the development of new structural materials with improved performance with wide applications in industry. They have excellent resistance to oxidation and sulfidation [1,2,3]. Their density is about two-thirds that of the steel density. The balance of the strength and plasticity of these materials is relatively good [4,5,6,7,8,9,10]. The input raw materials are relatively cheap due to their abundance in the Earth’s crust. Therefore, one may ask how far can these preferential properties (based on the B2 and DO_3_ structures) be maintained or even improved via additional alloying. Similarly, the main drawbacks of these alloys (bad workability at room temperature and low high-temperature (HT) strength) have eventually been improved.

The structural and mechanical properties of complex concentrated alloys (CCAs) are a widely studied subject [11,12,13,14,15,16,17,18,19,20,21,22,23,24,25,26,27]. Various single additives—Ti (0.5 to 4 at. %), Cr (0.5 to 8 at. %), Mo (0.5 to 4 at. %), and V (0.5 to 8 at. %)—effectively improved the yield stress of Fe_3_Al at 1073 K (800 °C) [11]. The effect of higher concentrations of Cr and V (up to 25 at. %) on the structural and high-temperature mechanical properties was also studied [12]. A detailed study of the ordering in highly alloyed (Cr, V) iron aluminide FeAl using the ALCHEMI method [27] showed that Cr atoms preferentially occupy the positions of Fe atoms and V those of Al. With increasing concentrations of Cr and V, the B2 order decreases, leading to full disorder (A2 structure) in the equiatomic FeAlCrV.

Adding the fifth element gets us to the realm of high-entropy alloys (HEAs). HEAs with body-centered cubic (bcc) structures possess excellent strength, but they often suffer from limited plasticity at room temperatures. Nevertheless, there are some promising results: Wu et al. [15] studied structure development and mechanical properties of TiZrNbMo_x_V_y_ alloy dependent on the content of Mo (x = 0–2) and V (y = 1 and 0.3). It is shown that the HEAs with low content of V are composed of only one type of bcc solid solution phase and demonstrate excellent phase stability at 1273 K. The high content of V and Mo results in the formation of two types of bcc solid solution phase and a decrease in phase stability in the HEAs. On the other hand, the HEAs with face-centered cubic (fcc) structures have sufficient plasticity, but the strength at high temperatures is substantially lower than for materials with bcc structures [25,26]. He et al. studied a series of six-component (FeCoNiCrMn)_100−x_Al_x_ (x = 0–20 at. %) HEAs. It was found that the crystalline structure changed from the initial single fcc structure to a duplex fcc+bcc structure and then a single bcc structure as the Al concentration increased. Structural changes were accompanied by corresponding variations in tensile properties. In the single fcc region, alloys behaved like a solid solution with relatively low strength but extended ductility [26].

In our previous study [12], eight iron aluminide alloys with different contents of V and Cr were prepared up to 25 at. % of both elements. All the alloys were in a solid solution condition without any major chemical inhomogeneity. For all alloys, a comparable grain size and D0_3_ crystallographic structure was observed. In the present work, we want to extend the knowledge about the structure of CCAs based on Fe_3_Al for the case of simultaneous addition of Cr, V, and Ni. It will be interesting to see how far will the D0_3_ structure of parent Fe_3_Al be retained under these conditions. Moreover, structural information alone can be used to verify and possibly improve material modeling within the Calphad approach.

## 2. Materials and Methods

A set of alloys based on Fe_3_Al iron aluminide was prepared. The ratio of Fe:Al was kept constant (3:1) while nominal concentrations of Ni, Cr and V varied: 8, 12, 15, and 20 at. %. These alloys were denoted as 50-F3A, 51-F3A, 54-F3A and 52-F3A, respectively. The mean concentration of the technological impurities coming from the metals used for the preparation of the alloys was: 0.01 at. % B, 0.1 at. % Mn and 0.06 at. % C. The volume fraction of such particles (mainly V carbide and Al oxide) is smaller than 1 vol.%.

The ingots were cast under a vacuum using an induction furnace with intensive stirring during melting. The subsequent slow cooling took place in the argon atmosphere. All alloys were further annealed at 1000 °C for 10 days in order to prepare homogeneous material in thermodynamic equilibrium.

A scanning electron microscope (SEM), Zeiss Auriga Compact, was used to visualize the structure of the alloys. The microscope was equipped with an EDAX detector for energy dispersive X-ray analysis (EDX). The surface of the alloys was prepared via mechanical polishing with a decreasing grain size down to 25 nm. The chemical composition of all samples was measured from a sufficiently large area—larger than 2 × 2 mm^2^. At least three measurements were performed for each sample.

Transmission electron microscopy (TEM) observations were carried out on JEOL JEM-2200FS operating at 200 kV. Samples were polished on SiC down to the thickness of about 100 µm. Finally, Struers Tenupol 5 was used to electropolish the specimens at 15 V and −25 °C in the 17% solution of HClO_4_ in methanol. Electropolishing in 20% solution of HNO_3_ in methanol—an usual electrolyte for FeAl alloys—resulted in preferential etching of the bcc phase.

X-ray diffraction (XRD) was performed using a Rigaku Rapid II diffractometer with a sealed Mo tube, curved 2D detector for measurement in transmission mode and a Mo Kα graphite monochromator (λ = 0.0709319 nm). Diffraction patterns from mechanically thinned TEM samples were acquired while rocking in two axes.

## 3. Results and Discussion

X-ray diffraction (Figure 1) revealed the bcc phase and its ordered variants B2 and D0_3_. Additionally, the alloy with the highest content of additional elements also contained the so-called σ-phase with a tetragonal structure. D0_3_ order was found in 50-F3A with a hint of a (111) D0_3_ peak also present in 51-F3A. B2 and bcc peaks were present in all four alloys. The lattice parameters are reported in Table 1. Peak positions were determined by fitting the peak profile with a suitable function. Only prominent and well-separated peaks were used. The error of the bcc lattice parameters was taken as the standard deviation of the values determined from different peaks. The error of the lattice parameters of the σ-phase was determined via statistical bootstrapping (standard deviation of values determined from random subsets of peaks). The lattice parameters of the bcc-based phases are close to that of the bcc elements (Fe 0.2886 nm [28], Cr 0.28839 nm [28] and V 0.30338 nm [29]), B2 compounds (NiAl 0.28812 nm [30], FeAl 0.29022 nm [31]) and Fe_3_Al (2 × 0.28945 = 0.5789 nm [31]). σ-phase with similar lattice parameters is found in Fe-Cr (a = 0.87995 nm, c = 0.45442 [32]), Fe-V (a = 0.8956 nm, c = 0.4627 nm [33]) and Ni-V (a = 0.898 nm, c = 0.464 nm [34]) binary systems.

The chemical composition of the studied alloys is presented in Table 1. Figure 2 shows an overview of the back-scattered electron (BSE) images of alloys 51, 54 and 52-F3A. No phase contrast (only impurities) was observed in alloy 50-F3A.

TEM observations of 50-F3A revealed single-phase material (except for a few Al oxide inclusions) with D0_3_ structure and nanometer domain size (Figure 3).

The microstructures of 51-F3A and 54-F3A were similar: both alloys show the separation of bcc and B2 on two length scales. There are alternating areas of bcc and B2 structure a few micrometers in size. These are further filled by much smaller (5–50 nm) particles of opposite phase, B2 and bcc, respectively (Figure 4 and Figure 5). The structure of 51-F3A appears more “blocky” without clear particle/matrix distinction, while B2 areas in 54-F3A seem to form rounded particles in the bcc matrix. Similarly, both B2 and bcc nanoscale particles in 54-F3A are round, while the nanoscale bcc phase in 51-F3A seems to form interconnected channels. The insides of the bigger-than-nanoscale B2 particles in both 51-F3A and 54-F3A show signs of further B2/bcc separation. Everything is crystallographically coherent. The weak (111) D0_3_ spot is visible in the 51-F3A [110] diffraction pattern, while in 54-F3A, it is even weaker and diffused. Figure 4d (dark field from (111) D0_3_) shows the nanometer D0_3_ phase, presumably on bcc/B2 interfaces in 51-F3A (that are present everywhere). In 54-F3A, the (111) D0_3_ intensity was too weak for a dark field image.

Alloy 52-F3A is composed of two phases: B2 and tetragonal σ-phase. They form distinct grains a few micrometers in size. Moreover, some grains of σ with included particles of B2 were also observed (Figure 6). In the latter case, there is a fixed orientation relationship between σ and B2. The independently determined lattice of the σ phase matches the structure identified via XRD, though hints of slight deviations from orthogonality at the limit of measurement accuracy (<1°) were noticed.

Figure 7 shows the EDX maps of the investigated alloys. The B2 phase in all two-phase materials (51, 54 and 52-F3A) shows a high content of Ni and Al. In 52-F3A, there is only very little Al in the σ phase. Quantitative results are presented in Table 2. Detailed quantitative analysis of the phase composition was complicated due to the strong absorption of Al Kα in the L lines of the other elements. This was accounted for by scaling the raw counts using custom k-factors. These were chosen so that the overall alloy composition matched the composition determined via SEM (Table 1). The nanoscale phase separation in alloys 51 and 52 is another issue; although high magnification maps and principal component analysis (PCA) were used, there is probably still some mixing between the phases. The values in Table 2 may, thus, come short of real tie-line endpoints, but the tie-line direction should be correct.

Although the idea was to keep the stoichiometry of Fe_3_Al fixed in order to somehow “impose” the Fe_3_Al’s D0_3_ structure on a final alloy, while V, Cr and Ni were added, it did not quite work out (except for the most dilute 50-F3A). Ni binding to Al is much stronger than the binding between Fe and Al (heat of formation NiAl 0.65 eV/atom vs. FeAl 0.3 eV/atom [35]). Increasing amounts of Ni thus consume more and more Al to form B2 NiAl. In 52-F3A, there is no more Al left in the rest of the alloy and σ phase forms.

To further discuss the phase composition in this system, a phase diagram section (isopleth) containing the nominal compositions of studied alloys was calculated (Figure 8a,b). ThermoCalc (version 2022a) software with the TCHEA5 database [36,37] was used. The calculated composition of tie-line endpoints (phases) is reported along with experimental results in Table 2.

The D0_3_ order is not modeled in this database. However, it should only become relevant at much lower temperatures than those depicted. The D0_3_ order observed in some of our alloys thus probably does not reflect the annealing equilibrium at 1000 °C but was formed after quenching. This is consistent with its nanometer length scale.

As can be seen from Figure 8a, the σ phase is incorrectly predicted for 51 and 54-F3A. Figure 8c depicts the Gibbs energy differences between the relevant phase combinations. In order to better describe the phase relations in this system, we tried to “fix” the TCHEA5 description of the σ phase by artificially increasing its Gibbs energy by 1.5 kJ/mol (45 kJ per mole of formula unit using PHASE_ADDITION command). This shifts the B2′+σ above B2′+bcc (B2′+B2) in 54-F3A (51-F3A) while still keeping it stable in 52-F3A. The modified phase diagram in Figure 8b is more consistent with our present observations. Except for a difference in ordering in 51-F3A, this diagram now correctly reproduces the observed phase relations. Despite the experimental challenges, the match between the experimental and calculated phases= compositions (Table 2) is also rather good.

## 4. Conclusions

The structure of a series of CCAs derived from the iron aluminide Fe_3_Al alloyed by Cr, Ni and V was determined. The observations were compared to the phase compositions predicted using the CalPhad approach.

Fe_3_Al maintains the D0_3_ structure up to at least 24 at. % of combined alloying with V, Cr and Ni.Beyond that (36 and 45 at. % of combined alloying by V, Cr and Ni), the alloys are composed of B2 NiAl together with bcc solid solution. Coherent phase separation occurs on two length scales down to a few nanometers.After the amount of alloyed Ni exceeds the amount of Al present in the alloy (the total combined alloying by V, Cr and Ni of 60 at. %), the alloy forms FeNiCrV σ-phase together with B2 NiAl.The original TCHEA5 database overestimates the stability of the σ phase. The present experimental results can thus serve to its improvement.

## Figures and Tables

**Figure 1 materials-16-05388-f001:**
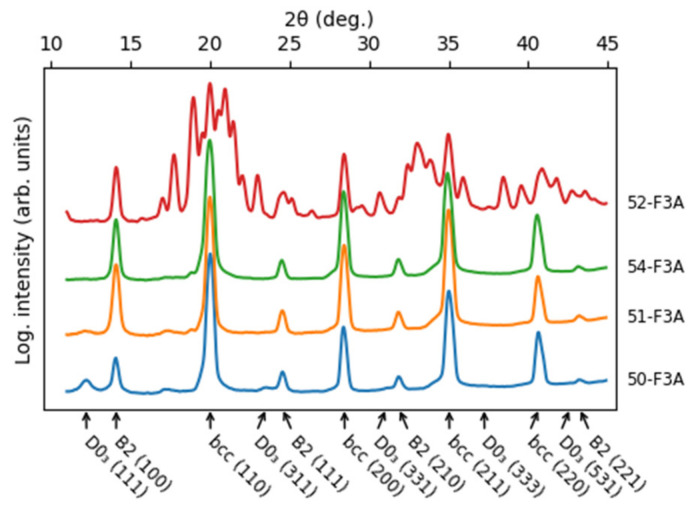
X-ray diffraction patterns from studied alloys. Positions and indices of peaks of bcc-based phases are indicated. Additional peaks in 52-F3A belong to σ-phase.

**Figure 2 materials-16-05388-f002:**
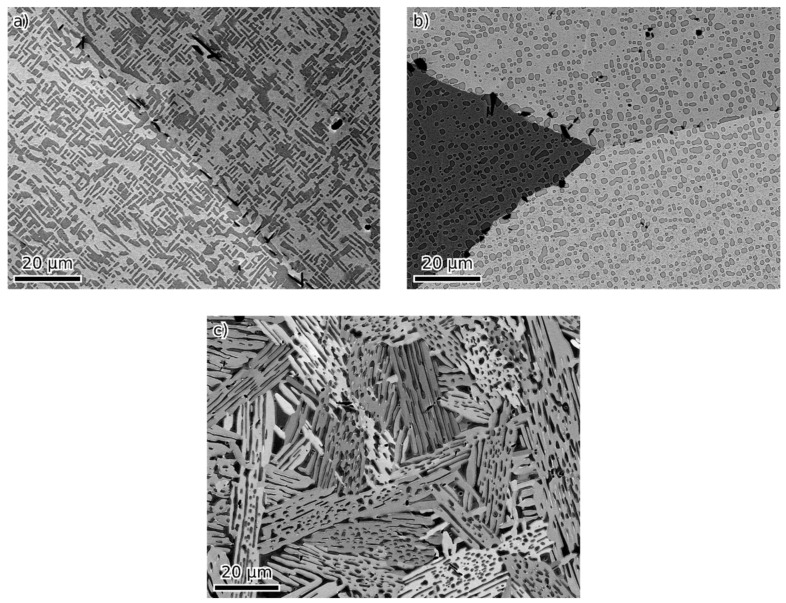
SEM BSE images of alloys 51, 54 and 52-F3A (**a**), (**b**) and (**c**), respectively.

**Figure 3 materials-16-05388-f003:**
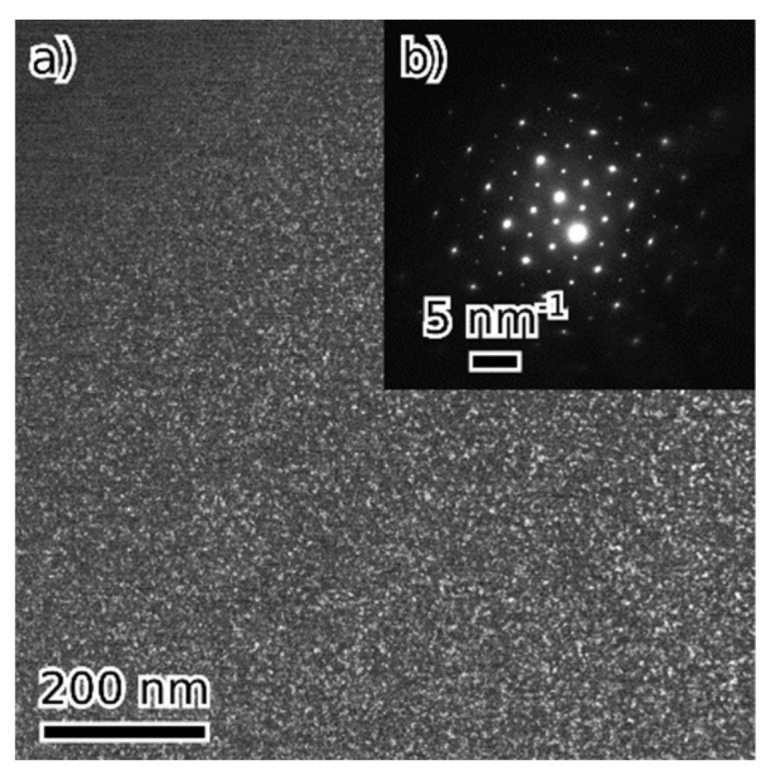
Single-phase 50-F3A with nanoscale D0_3_ domains: dark field image from (111) D0_3_ diffraction spot (**a**), [110] diffraction pattern (**b**).

**Figure 4 materials-16-05388-f004:**
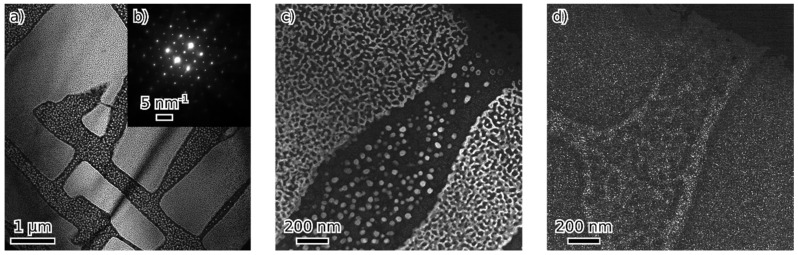
Alloy 51-F3A: dark field image from (100) B2 spot (**a**,**c**), bcc phase is dark, [110] diffraction pattern (**b**), microstructure detail (**c**) and a dark field image from (111) D0_3_ spot (**d**).

**Figure 5 materials-16-05388-f005:**
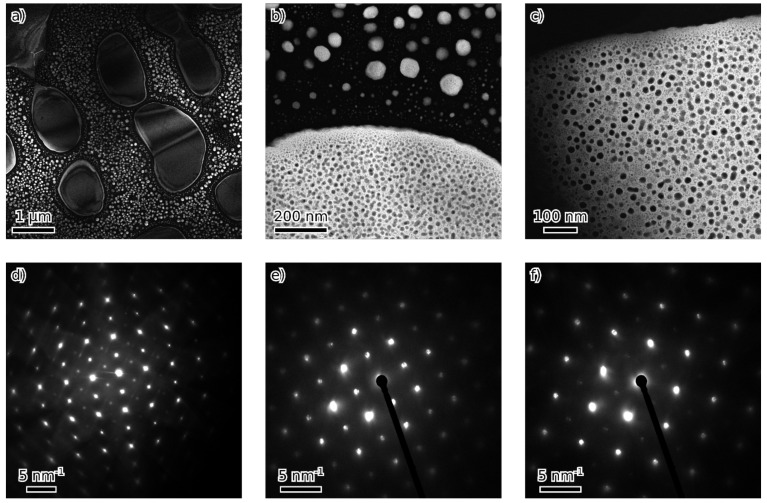
Alloy 54-F3A: dark field images from (100) B2 spot at different levels of detail (**a**–**c**), bcc phase is dark, [110] diffraction pattern (**d**). [100] nanodiffraction from B2 (**e**) and bcc (**f**) areas.

**Figure 6 materials-16-05388-f006:**
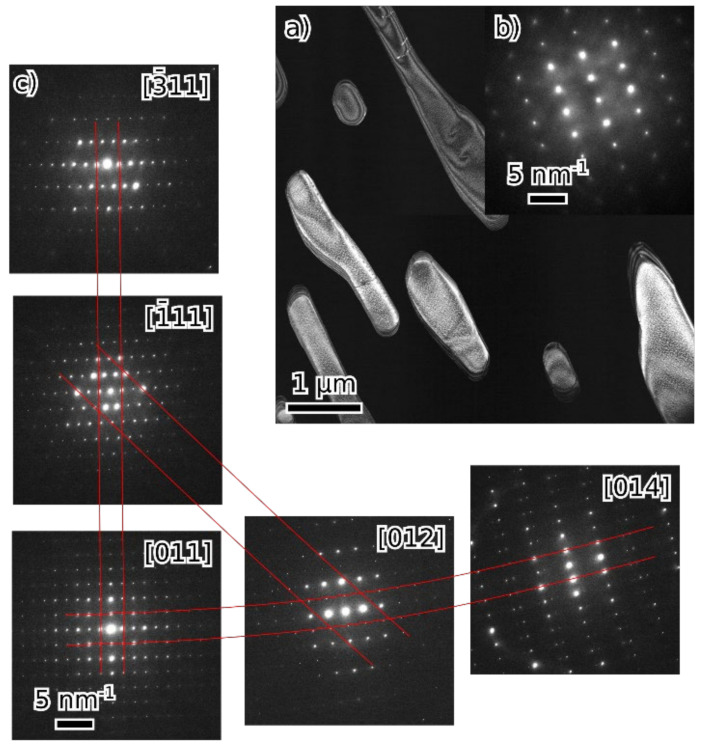
Alloy 52-F3A: dark field image showing B2 particles in a single grain of σ-phase ((**a**), σ is dark), [110] B2 diffraction pattern (**b**) and several low index σ diffraction patterns (**c**) from single a grain with a schematic depiction of their arrangement and lines highlighting common planes.

**Figure 7 materials-16-05388-f007:**
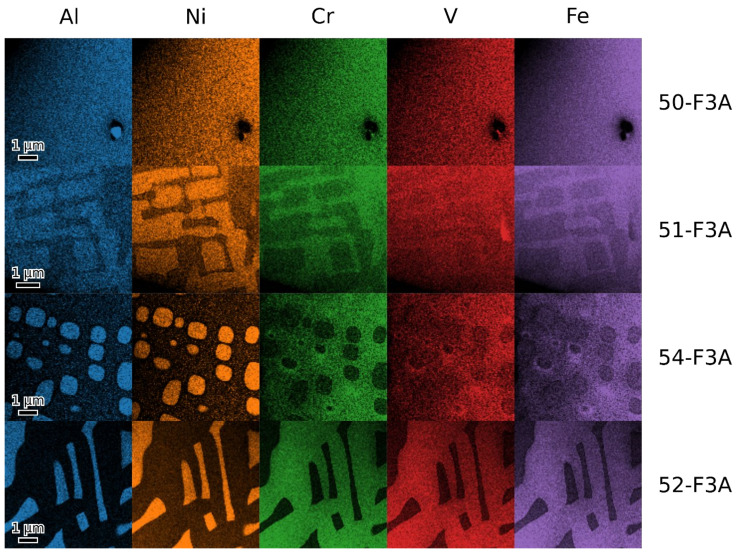
EDX composition maps of alloys 50, 51, 54 and 52-F3A shown in respective rows.

**Figure 8 materials-16-05388-f008:**
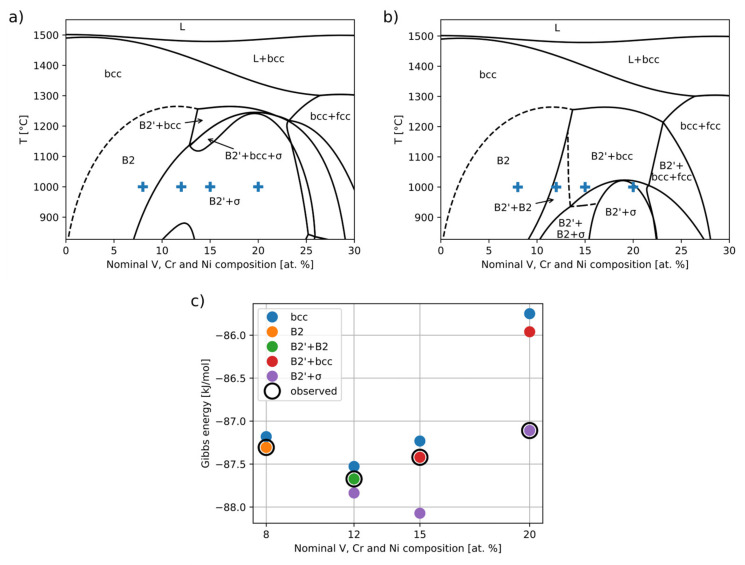
Isopleth phase diagram calculated from original TCHEA5 data (**a**) and with modified (see main text) description of σ phase (**b**). Composition and annealing temperature of studied phases are indicated by crosses. Plot (**c**) shows relative phase stability of relevant phase combinations. B2 and B2′ represent two distinct B2 phases corresponding approximately to FeAl and NiAl, respectively.

**Table 1 materials-16-05388-t001:** Overall composition of studied alloys determined via EDX analysis together with an overview of phases identified using TEM and lattice parameters determined via XRD (½ of D0_3_ lattice is reported for 50-F3A).

at. %	Fe	Al	Ni	V	Cr	Phases	Lattice Parameters (nm)
50-F3A	56.8	19.1	8.1	8.0	8.5	D0_3_	a = 0.2890(3)
51-F3A	46.5	18.0	11.5	11.5	11.6	bcc + B2 + D0_3_	a = 0.2890(1)
54-F3A	40.1	14.4	15.5	15.1	14.8	bcc + B2 (+D0_3_)	a = 0.2893(2)
52-F3A	29.8	9.4	19.9	20.5	20.4	B2 + σ	a = 0.2888(1) a_σ_ = 0.8887(2) c_σ_ = 0.4605(2)

**Table 2 materials-16-05388-t002:** Composition of phases in two phase alloys determined by TEM EDX (exp.) and calculated via ThermoCalc (TC). B2 and B2′ represent two distinct B2 phases corresponding approximately to FeAl and NiAl, respectively.

at. % ± ~3 at. %			Al	V	Cr	Fe	Ni
51-F3A	exp.	bcc	13.6	12.6	13.5	51.7	8.5
		B2′	23.1	10.4	9.5	41.6	15.3
	TC	B2	15.1	12.4	12.5	49.2	10.9
		B2′	33.4	4.2	2.6	26.5	33.3
54-F3A	exp.	bcc	6.3	18.7	20.6	50.0	4.3
		B2′	26.0	9.8	6.0	25.9	32.3
	TC	bcc	9.8	17.1	17.4	45.4	10.3
		B2′	34.8	3.9	2.1	18.9	40.3
52-F3A	exp.	σ	4.0	24.0	24.9	34.1	12.9
		B2′	26.6	8.7	5.0	14.7	44.9
	TC	σ	0.02	26.3	27.3	37.9	8.5
		B2′	33.8	5.0	2.6	11.1	47.5

## Data Availability

The data sets generated during this study of material are available from the corresponding author upon reasonable request.

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
