# Peer review of "Structure of Complex Concentrated Alloys Derived from Iron Aluminide Fe3Al"

_materials, 2023, doi:10.3390/ma16155388_

Round 1
Reviewer 1 Report
comments are embedded in the manuscript. good paper!

Reviewer 2 Report
This is a well-written paper presenting novel results. Some comments are given below.
1. Authors found the alloy 52-F3A is composed of two phases: B2 and tetragonal σ-phase. It is different from 51-F3A and 54-F3A alloys. It is useful to explain their different formation mechanisms.
2. Authors indicated that detailed quantitative analysis of phase composition of alloys 51 and 54 is complicated. It is useful to provide a reasonable or estimated quantitative analysis for the comparison purpose.
3. No material properties were presented for different samples. It is difficult to realize the effect of crystal structure on the strength of the materials. Authors are suggested to provide explanations or additional experimental data to support this issue.
Reviewer 3 Report
1. It would be great if authors could add the data for Fe3 Al , Fe3Al Nix, Fe3Al Crx, and Fe3Al Vx in introduction part.
2. Why did authors use annealing? why they didnt test the as cast samples
3. authors should discuss more about formation of phases. especially formation of B2 and sigma phase, and effect of each phase in mechanical properties.
4. please calculate solid solution formation possibility
in current version, authors only provided results , without enough scientific discussion.
Round 2
Reviewer 3 Report
accepted